# Is Environmental Cadmium Exposure Causally Related to Diabetes and Obesity?

**DOI:** 10.3390/cells13010083

**Published:** 2023-12-30

**Authors:** Soisungwan Satarug

**Affiliations:** Kidney Disease Research Collaborative, Translational Research Institute, Woolloongabba, Brisbane, QLD 4102, Australia; sj.satarug@yahoo.com.au

**Keywords:** bilirubin, cadmium, diabetes type 2, glucose metabolism, heme oxygenase-1, heme oxygenase-2, obesity

## Abstract

Cadmium (Cd) is a pervasive toxic metal, present in most food types, cigarette smoke, and air. Most cells in the body will assimilate Cd, as its charge and ionic radius are similar to the essential metals, iron, zinc, and calcium (Fe, Zn, and Ca). Cd preferentially accumulates in the proximal tubular epithelium of the kidney, and is excreted in urine when these cells die. Thus, excretion of Cd reflects renal accumulation (body burden) and the current toxicity of Cd. The kidney is the only organ other than liver that produces and releases glucose into the circulation. Also, the kidney is responsible for filtration and the re-absorption of glucose. Cd is the least recognized diabetogenic substance although research performed in the 1980s demonstrated the diabetogenic effects of chronic oral Cd administration in neonatal rats. Approximately 10% of the global population are now living with diabetes and over 80% of these are overweight or obese. This association has fueled an intense search for any exogenous chemicals and lifestyle factors that could induce excessive weight gain. However, whilst epidemiological studies have clearly linked diabetes to Cd exposure, this appears to be independent of adiposity. This review highlights Cd exposure sources and levels associated with diabetes type 2 and the mechanisms by which Cd disrupts glucose metabolism. Special emphasis is on roles of the liver and kidney, and cellular stress responses and defenses, involving heme oxygenase-1 and -2 (HO-1 and HO-2). From heme degradation, both HO-1 and HO-2 release Fe, carbon monoxide, and a precursor substrate for producing a potent antioxidant, bilirubin. HO-2 appears to have also anti-diabetic and anti-obese actions. In old age, HO-2 deficient mice display a symptomatic spectrum of human diabetes, including hyperglycemia, insulin resistance, increased fat deposition, and hypertension.

## 1. Introduction

Cadmium (Cd) is a redox inert metal, present in relatively low levels in the Earth’s crust and most surface soils [1,2,3]. It is primary present as a sulphide, such as in greenockite and the zinc ore, sphalerite [2]. Hence, mining, smelting, and refining of zinc ores yield Cd as the main byproduct. It has been widely used in many industrial processes because of its metallic and anti-corrosive properties [1,2]. Health threats posed by this metal have been perceived because of the “itai-itai” disease found in the Jinzu river basin of Japan. The disease was found to be due to excessive exposure to Cd through consumption of rice grown on paddy soils contaminated with the discharge from zinc mining [4,5,6]. 

Because of its notoriously high toxicity, the worldwide production and industrial applications of Cd have significantly declined [3]. However, the use of Cd-contaminated phosphate fertilizers persists, and adds substantial amounts of Cd to the food chain in most parts of the world [7,8,9,10]. Like all other metals, Cd is nonbiodegradable, and consequently, it persists indefinitely in the environment, and it can accumulate in vegetation even the levels of Cd in soils are very low [6,11]. 

The kidney and bones were described as the organs severely affected in “itai-itai” disease patients [4,5]. Current evidence, however, suggests that the effects of Cd exposure extend beyond the kidney and bones. This is evident from epidemiological data that link hypertension, diabetes type 2 (DM), chronic kidney disease (CKD), osteoporosis, non-alcoholic fatty liver disease, infertility, and various types of cancer to environmental Cd exposure in the general populations of many countries [12,13,14]. 

This present review focused on the impact of environmental Cd exposure on the prevalence of pre-diabetes and diabetes, which are defined as fasting plasma glucose ≥ 110 mg/dL and 126 mg/dL, respectively [(https://www.cdc.gov/diabetes/basics/getting-tested.html) (accessed on 28 November 2023)]. The global prevalence of diabetes has now reached epidemic proportions, and the epidemic is most frequently attributed to the concurrently increasing prevalence of obesity [(https://www.who.int/health-topics/diabetes#tab=tab_1) (accessed on 25 December 2023). Environmental Cd exposure is unrecognized and ignored contributing factor.

The diabetogenic action of chronic exposure to Cd was first noted in 1980 by Merali and Singhal, using neonatal rats [15]. Prior to this landmark observation, experimental data indicate that Cd had the ability to disrupt hepatic and renal glucose metabolism that persisted one year after exposure was discontinued [16,17,18,19]. These findings underscore the indispensable roles of the kidney and liver in the maintenance of blood glucose levels [20,21,22,23,24].

This review has three major aims. Firstly, to define dietary sources of Cd from total diet studies together with exposure levels, reflected by blood and urinary Cd levels that may increase risks of prediabetes and diabetes in the environmentally exposed populations. Secondly, to evaluate if current exposure guidelines and the Cd nephrotoxicity threshold level are sufficiently low to provide health protection for at least 95% of the population. Thirdly, to explore mechanisms underlying diabetogenic effects of Cd and the strategies to mitigate the cytotoxicity of Cd through modulation of cellular stress response and defense mechanisms, mediated by heme oxygenase-1 and -2 (HO-1 and HO-2). 

## 2. Exposure Sources, Dosimetry and Health Risk Assessment 

### 2.1. Dietary Exposure to Cadmium 

Diet is the major environmental sources of Cd exposure in non-smokers and non-non-occupationally exposed persons. This exposure source is indicated by the presence of Cd in the human diet, reported in total diet study (TDS) [25,26]. TDS is a food safety monitoring program, known also as the “market basket survey”, conducted by food authority agencies [25,26]. Typically, samples of foodstuffs are collected from supermarkets and retail stores for measurement of food additives, pesticide residues, contaminants, and nutrients. The exposure level is calculated from the concentration of a given contaminant and an amount of the food item consumed each day. To approximate the exposure levels among average and high consumers, the median and the 90th percentile concentration levels of a contaminant are used, respectively [27,28]. 

Levels of Cd in the human diet appeared to vary widely among populations, but the major sources of dietary Cd are foods that are frequently consumed in large quantities, such as rice, potatoes, wheat, and leafy salad vegetables [14]. Of concern, rice is a staple food for over half of the world’s population, and in some regions, rice contributes to more than 50% of the total Cd intake, detailed below. 

In two Cd-polluted areas of Japan, dietary Cd exposure levels among women were 55.7 µg/d (1.03 µg/kg body weight) and 48.7 µg/d (0.86 µg/kg bw/d) [6]. Rice and its products constituted 40–50% of these Cd exposures. Differences between the two groups with respect to dietary Cd exposure were due mostly to Cd levels in rice consumed in the two locations. Cd was found in all plant foods investigated, especially in spinach, Japanese parsley, garland chrysanthemum, Japanese mustard spinach, belvedere fruit, shiitake mushrooms, and seaweed. Shellfish, salted squid guts, scallops, oysters, and freshwater clams were substantial animal sources of Cd. Chocolate and tea leaves contained also high-Cd levels. Cd was not found in brewed tea.

In the general Japanese population, the mean dietary Cd exposure was 0.35 μg/kg bw/day, ranging between 0.25 and 0.45 μg/kg bw/day [29]. Respective contribution of exposure to Cd in rice and its products, green vegetables, cereals, and seeds plus potatoes were 38%, 17%, and 11%. 

Dietary Cd exposure in China was 32.7 μg/day, with rice and vegetables being the main sources [30]. Potatoes were the main dietary Cd source in Mongolia, and it contributed to 24% of total Cd exposure [30]. Notably, high Cd levels were found in Nori, peanuts, squid, cuttlefish, and mushrooms [31,32].

Dietary Cd exposure in South Korea was 12.6 μg/day [33] with cereals and vegetables, beverages, fruits and nuts, and dairy products (milk included) being the main sources. Relatively high Cd concentrations were found in cereals, oil seeds and fruits, and vegetables. Another Korean study estimated Cd exposure of 22 μg/day [34]. Average rice consumption ranged between 587 and 611 g/day, and it was the major contributor to total Cd exposure (40.3%), followed by squid (11.8%), eel (11.0%), crab (8.6%), shellfish (3.6%), kimchi (Korean cabbage; 3.5%), and seaweed (3.5%).

### 2.2. The Intestinal Absorption of Cadmium

For most people, exposure to Cd is unavoidable because it is present in nearly all food types. The living organism does not synthesize nor break down metals, and transporter systems and pathways have evolved, consequently, to acquire from exogenous sources all metals [35,36,37,38,39]. These multiple transporters systems, in turn, provide Cd entry routes into most cells in the body. In all likelihoods, Cd in the gut gains an entry into the portal blood system through transporters for calcium, zinc, manganese, iron, copper, and cobalt (Ca, Zn, Mn, Fe, Cu, and Co) [40,41,42]. Examples of such metal transporters are members of the Zrt- and Irt-related protein (ZIP) of the zinc transporter family and the Ca^2+^-selective channel TRPV6 [43,44,45,46,47]. Furthermore, Cd complexed with the metal binding protein, metallothionine (MT) and phytochelatin (PC) as CdMT and CdPC can be absorbed through transcytosis, and endocytosis, mediated by the human neutrophil gelatinase-associated lipocalin (hNGAL) [48,49,50].

Similarly, through the transporters for essential metals, Cd can enter most cells in the body, including hepatocytes [51], kidney tubular epithelial cells [52,53,54,55,56,57], adipocytes [58], insulin producing pancreatic β-cells [59], ovaries [60], testes [61], and erythrocytes [62,63,64]. However, no physiologic mechanism exists to eliminate Cd. Virtually all acquired metal is consequently retained, and the cellular levels of Cd increased with age (duration of exposure). 

Only a miniscule amount of Cd (0.001–0.005% of the body burden) is excreted each day. Consequently, the body burden of Cd is essentially determined by intestinal absorption rate. In theory, the absorption rate of Cd will increase when the body content of essential metals; Fe, Zn, Ca is low, and when diets are deficient in these nutritionally essential metals.

Table 1 provides Cd accumulation levels in various tissues, recorded in Australian and Japanese autopsy studies.

Preferential Cd accumulation in the kidney cortex in the female gender were apparent from both Australian and Japanese autopsy studies [51,52]. On average, the hepatic Cd level in Australian women was 1.74-fold higher than men, and after adjustment for inhalation exposure, women had a higher kidney cortical Cd content than men did [51]. In comparison, the mean liver Cd in women living in a non-Cd contaminated location of Japan was 1.6-fold higher than men [52]. 

Fractionally, the difference between men and women in kidney cortex content is smaller than the difference in hepatic content. A plausible interpretation is that women have lower iron stores, and adjustments to increase intestinal iron absorption lead to increased absorption and liver uptake of Cd from dietary exposure. Redistribution of hepatic Cd to the kidney may be sufficient to cause a higher kidney content of Cd as well, but not so great as to obscure the dietary origin of the increased Cd burden.

### 2.3. Blood Cadmium

Cd enters erythrocytes through the chloride/bicarbonate anion exchanger ([Cl^−^/HCO^3−^], AE1, SLC4A1) [61,62,63], and iron transporters that were responsible also for erythrocytic uptake of lead (Pb) and Zn [65,66,67]. Cd may induce the erythrocyte membrane morphological change [68], leading to premature hemolysis in the reticuloendothelial system, and thereby shorten cellular lifespan [69]. Cd may induce eryptosis, erythrocytic suicidal cell death, which is the mechanism to eliminate injured red blood cells [70,71]. 

Most of the circulating Cd is bound to hemoglobin in red blood cells [72,73,74,75,76]. Less than 10% of Cd the circulation is in the plasma, where it is associated with histidine and thiols (-SH) of peptides and proteins, such as pre-albumin, albumin, α_2_-macroglobulin, and immunoglobulins G and A [74,77,78]. Examples of the non-protein plasma thiols that interact with Cd are glutathione (GSH), cysteine, cysteinylglycine, homocysteine, and γ-glutamylcysteine [79,80]. The total concentrations of these non-protein thiols are in the low µM range (12–20 µM). In comparison, albumin thiol is more abundant (0.6 mM), implying a significant role of albumin in the transport and delivery of Cd to cells throughout the body [81].

The estimated half-life of blood Cd varied from 75 to 128 days [82]. Because the mean life-span of erythrocytes is 120 days, blood Cd is used as an indicator of recent exposure to the metal. In an epidemiologic investigation, a significant correlation was observed between blood and urine Cd, which suggested that a blood Cd level may reflect partially long-term exposure [83]. 

In theory, plasma Cd is more predictive of tissue toxic injury than erythrocytic Cd because plasma Cd is readily exchangeable with other metals in target tissues. However, use of plasma Cd in exposure assessment is limited because of the high cost involved in its quantification as plasma Cd is in a nano molar (nM) range. Presently, the distribution of Cd in whole blood and plasma remains to be determined. A more precise health risk assessment could be made if the relationship between blood and plasma Cd at varying exposure levels is known.

### 2.4. Excretion of Cadmium Siginfies Kidney Tubular Cell Injury and Death 

The kidney is an organ where the most Cd accumulates because kidney tubular epithelial cells are responsible for the reabsorption of virtually all proteins in the ultrafiltrate [84,85,86,87]. In addition, there are various metal transporters expressed in the apical and basolateral membranes of tubular cells [14]. These metal transporters and protein re-absorptive pathways provide Cd several entry routes into tubular cells along with proteins to which it is bound such as albumin, transferrin (Tsf) [87]. 

Experimental studies in rats using microinjection technique found that 70–90% of Cd was taken up in the S1-segment of the proximal tubule [88,89], where the megalin/cubillin receptor-mediated endocytosis is involved. The neutrophil gelatinase-associated lipocalin (NGAL)/lipocalin-2 receptor system has also been implicated in reabsorption of Cd-protein complexes in the distal tubule and the collecting duct [56,90,91]. 

There are multiple entry routes of Cd but there is no exit route, which means that most acquired Cd is retained in kidneys and is released to tubular lumen when cells die due to the toxicity of Cd accumulation [14,92]. Thus, the excretion of Cd signifies tubular cell injury and death induced by a cumulative burden of Cd [14,92]. Figure 1 shows the mitochondrion as the toxicity target of Cd.

Through various metal transporters, Cd reaches the inner membrane of the mitochondria, where it affects the synthesis of adenosine triphosphate (ATP), inhibits the electron transport chain, and promotes the formation of reactive oxygen species (ROS) with resultant oxidative stress conditions [57,93,94,95]. 

The organs with high metabolic activities and high energy demands, like the kidneys, ovaries, and testes are particularly sensitive to Cd-induced mitochondrial dysfunction, a central mechanism by which Cd affects most cells in the body [14]. 

### 2.5. Is Urinary β_2_M Indicative of Tubulopathy? 

β_2_M protein with a molecular weight of 11,800 Da, is synthesized and shed by all nucleated cells [96]. β_2_M undergoes filtration and all filtered β_2_M is reabsorbed by proximal tubular cells [97]. Increased β_2_M excretion has been used as indicator of tubulopathy [98,99] and was described as a dominant feature of Cd nephropathy. However, some attributes of β_2_M excretion compromise its utility for such purposes. First, β_2_M production rises in response to many inflammatory and neoplastic conditions [100]. Second, if reabsorption rates of β_2_M per nephron remain constant as its production rates change, excretion will vary directly with its production. Third, if the production and reabsorption per nephron remain constant as nephrons are lost, the excretion of β_2_M will rise.

The increased β_2_M excretion due to Cd-induced nephron loss has been revealed in a dose–response analysis, where β_2_M excretion of 100–299, 300–999, and ≥1000 μg/g creatinine were associated with 4.7-fold, 6.2-fold and 10.5-fold increases in the likelihood of eGFR ≤ 60 mL/min/1.73 m^2^, indicative of substantial nephron loss [101]. 

A threshold of toxicity is defined as the highest dose that does not produce an adverse effect in the most sensitive organ (endpoint) [102]. A rise of β_2_M excretion above 300 µg/g creatinine (tubular proteinuria) is the manifestation of severe toxicity of Cd in kidneys, and its use as an endpoint to determine an exposure guideline is inappropriate.

Persistent toxicity from existing renal stores may eventuate in progression of CKD [103,104,105,106]. Current evidence suggests that Cd may impair tubular protein reabsorption by the receptor-mediated endocytosis (RME) as depicted in Figure 2 [76].

### 2.6. Health Risk Assessment of Cadmium Exposure 

#### 2.6.1. Exposure Guideline

The Joint FAO/WHO Expert Committee on Food Additives and Contaminants (JECFA) used tubular proteinuria, defined as a rise of urinary excretion of β_2_-microglobulin (β_2_M) above 300 µg/g creatinine, to indicate the nephrotoxicity of dietary Cd exposure [107]. Based solely on this endpoint, a tolerable monthly intake (TMI) of Cd was found to be 25 μg per kg body weight per month, equivalent to 0.83 μg per kg body weight per day [107]. A Cd excretion of 5.24 μg/g creatinine was suggested to be a nephrotoxicity threshold value [107]. 

The European Food Safety Authority (EFSA) employed the same endpoint, but a Cd excretion of 1 μg/g creatinine was designated as the toxicity threshold after inclusion of an uncertainty factor (safety margin) [108]. A dietary exposure of Cd at 0.36 μg/kg body weight per day for 50 years was viewed as an acceptable Cd ingestion level or reference dose (RfD) [108,109].

#### 2.6.2. Population Data

In a risk analysis of Chinese population data, a dietary exposure level of 16.8 µg/day for a 60 kg person (0.28 μg/kg body weight per day) was suggested to be a tolerable intake level, when tubular proteinuria (β_2_M) was an endpoint [110]. A corresponding threshold level of Cd excretion was 3.07 μg/g creatinine.

In a risk analysis of Thai population data, nephron loss was used as an endpoint, and Cd excretion level that is likely to produce a negligible adverse effect, termed benchmark dose limit or the NOAEL equivalent was 0.01 µg/g creatinine [111]. 

The benchmark dose limit (NOAEL equivalent) values have been calculated from different endpoints, including, albuminuria [112,113], enzyme-uria [114], diabetes [115], infertility [13], and bone loss [116,117,118]. These values all indicated that the toxicity of Cd occurs at very low body burden. 

In summary, a dietary exposure guideline of 0.83 μg per kg body weight per day (58 µg/day for a 70 kg person), and a nephrotoxicity threshold level of 5.24 µg/g creatinine [35,36] were established by the WHO to provide a safeguard against excessive exposure. These values were based solely on the excretion rate of β_2_M above 300 µg/g creatinine. However, population data, summarized in Table 2 (Section 3) show an association between an increase in risk of diabetes, and urinary Cd excretion five-to ten-fold below that the established Cd toxicity threshold level. This raises a serious concern that the current health guidelines do not afford health protection. 

## 3. Cadmium, Obesity, and Diseases with High Prevalence 

Numerous population studies have linked diseases with high prevalence, such as DM and CKD, to lifelong exposure to environmental Cd. In the present review, however, data from the U.S. general population, recorded in the National Health and Nutrition Examination Survey (NHANES) are highlighted. The U.S. NHANES provides data on levels of exposure to more than 200 chemicals, experienced by the representative of U.S. general population [119]. Urinary and blood Cd levels were quantified using standardized methodology that enables the comparison of data across NHANES cycles [119]. 

Table 2 provides evidence that Cd exposure, even at low levels, may increase the prevalence of pre-diabetes, diabetes, CKD and liver disease in the representative U.S. population [120,121,122,123,124,125,126,127,128,129,130]. 

**Table 2 cells-13-00083-t002:** Urinary and blood cadmium levels associated with increased risks of liver and kidney diseases in the United States.

NHANES	Exposure and Risk Estimates	References
1988–1994 n 8722, ≥40 years	Urinary Cd levels 1–2 μg/g creatinine were associated with prediabetes (OR 1.48) and diabetes (OR 1.24). Urine Cd levels > 2 µg/g creatinine were associated with 2.5-fold and 1.45-fold increases in risk of prediabetes and diabetes, respectively.	Schwartz et al., 2003 [120]
2005–2010 n 2398, ≥40 years	Urinary Cd > 1.4 µg/g creatinine in non-smokers were associated with pre-diabetes.	Wallia et al., 2014 [121]
2007–2012 n 3552, ≥20 years	Urinary Cd quartile 4 was associated with prediabetes among men (OR 1.95). OR for prediabetes rose 3.4-fold in men with obesity and a high Cd exposure, compared to those with a normal weight and low Cd exposure.	Jiang et al., 2018 [122]
1988–1994 n 12,732, ≥20 years	Urinary Cd levels ≥ 0.83 μg/g creatinine in women were associated with liver inflammation (OR 1.26). Urinary Cd ≥ 0.65 μg/g creatinine in men were associated with liver inflammation (OR 2.21), NAFLD (OR 1.30), and NASH (OR 1.95).	Hyder et al., 2013 [123]
1999–2015 n 11, 838, ≥20 years	A 10-fold increment of urinary Cd was associated with elevated plasma levels of ALT (OR 1.36), and AST (OR 1.31).	Hong et al., 2021 [124]
1999–2016 n 4411 adolescents	Urinary Cd quartile 4 was associated with elevated plasma ALT (OR 1.40) and AST (OR 1.64). The effect was larger in boys than girls.	Xu et al., 2022 [125]
1999–2006 n 5426, aged ≥20 years,	Urinary Cd levels ≥ 1 µg/L were associated with increased risk of albuminuria ^a^ (OR 1.41) and low GFR ^b^ (OR 1.48).	Ferraro et al., 2010 [126]
2009–2012, n 2926, aged ≥20 years	Urinary Cd levels > 0.220 μg/L were associated with increased albumin excretion, compared with urinary Cd < 0.126 μg/L. Blood Cd levels > 0.349 μg/L associated with increased albumin excretion, compared with blood Cd < 0.243 μg/L.	Zhu et al., 2019 [127]
2011–2012 n 1545, aged ≥20 years	Blood Cd levels > 0.53 μg/L were associated with albuminuria (OR 2.04) and low GFR (OR 2.21). OR for albuminuria was increased to 3.38 in those with similar Cd exposure levels and serum Zn < 74 μg/dL.	Lin et al., 2014 [128]
2007–2012 n 12,577, aged ≥20 years	Blood Cd levels > 0.61 μg/L were associated with low GFR (OR 1.80) and albuminuria (OR 1.60). GFR reduction associated with Cd was more pronounced in those with diabetes, hypertension, or both.	Madrigal et al., 2019 [129]
1999–2016, n 46,748, aged ≥20 years	Of 262 chemicals tested, blood Cd was associated with all three kidney outcomes; low GFR, albuminuria, and low GFR plus albuminuria.	Lee et al., 2020 [130]

NHANES, National Health and Nutrition Examination Survey; n, sample size; OR, odds ratio; NAFLD, non-alcoholic fatty liver disease; NASH, non-alcoholic steatohepatitis; ALT, alanine aminotransferase; AST, aspartate aminotransferase; ^a^ Albuminuria was defined as urinary albumin-to-creatinine ratio ≥ 30 mg/g creatinine in women and ≥20 mg/g creatinine in men; ^b^ Low GFR was defined as the estimated glomerular filtration rate < 60 mL/min/1.73 m^2^.

As data in Table 2 indicate, low environmental Cd exposure in the U.S. has been linked to CKD and liver disease in additional to prediabetes and diabetes.

Increases in the risks of prediabetes and diabetes among NHANES 1988–1994 participants were associated with urinary Cd levels of 1–2 µg/g creatinine [120]. An increased risk of prediabetes among NHANES 2005–2010 was associated with urinary Cd levels ≥ 0.7 µg/g creatinine after adjustment for covariates [121]. Obesity was associated with prediabetes in both men and women, and there was evidence that obesity may potentiate diabetogenic effects of Cd among men [122]. 

Of concern, evidence for hepatic effects of Cd has been found in both adolescents [125] and adults [123,124]. A study in rats provided evidence that diabetes may have an adverse effect on the liver [131].

### 3.1. Dietary Exposure: The U.S. Experience 

TDS data indicate average dietary Cd exposure in the U.S. was 4.63 μg/d [132]. This figure was computed from levels of Cd found in 260 food items in the 2006–2013 market basket survey together with 24 h dietary recall data from 12,523 participants in NHANES 2007–2012, aged 2 years and older. A dietary assessment of U.S. women (*n* = 1002, mean age 63.4) reported mean dietary Cd exposure of 10.4 μg/day, and mean Cd excretion of 0.62 μg/g creatinine [133].

Cereals and bread, leafy vegetables, potatoes, legumes and nuts, stem/root vegetables, and fruits contributed to 34%, 20%, 11%, 7%, and 6% of total intake, respectively. Foods that contain relatively high Cd levels are spaghetti, bread, potatoes, and potato chips which contributed the most to total Cd intake, followed by lettuce, spinach, tomatoes, and beer. Lettuce was a main Cd source for White people and Black people. Tortillas and rice were the main Cd sources for Hispanic Americans and Asians plus other ethnicities [132].

### 3.2. Cadmium and Its Inverse Relationship with Obesity 

Studies from the U.S. and other countries consistently observed inverse relationships of urinary and blood Cd levels with various measurements of adiposity, including increases in BMI, hip girth, and waist circumference. The reason for this phenomenon has never been investigated and largely ignored. However, it at least indicates that the diabetogenicity of Cd is unrelated to obesity, and Cd exposure at least accounts for diabetes among lean subjects. In a study from Uganda, 3 in 4 adults with diabetes were lean [134].

#### 3.2.1. Children and Adolescents 

Urinary Cd was associated with a reduction in risk of obesity in children and adolescents enrolled in NHANES 1999–2011 by 54%; an inverse association between obesity and urinary Cd was stronger in a younger (6–12 years) than the older age group (13–19 years) [135]. Both height and body mass index (BMI) in Flemish children, aged 14–15 years showed an inverse association with Cd excretion [136]. 

#### 3.2.2. Adults

Central obesity among participants of NHANES 1999–2002 inversely associated with Cd excretion [137]. Among NHANES 2003–2010 participants, their blood Cd levels inversely associated with BMI [138]. In another analysis of data from NHANES 2001–2014 participants aged 20–80 years (*n* = 3982), Cd excretion levels were not associated with the risk of metabolic syndrome, but they were associated with a reduced risk of abdominal obesity [139]. A meta-analysis of data from 11 cross-sectional studies indicated that Cd exposure was not associated with an increased risk of metabolic syndrome, but it was associated with dyslipidemia, especially in Asian population [140].

Similarly, an inverse association between BMI and blood Cd was seen in non-smokers in the Canadian Health Survey 2007–2011 [141]. In a study of the indigenous population of Northern Québec, Canada, where obesity was highly prevalent, an inverse relationship between Cd exposure and obesity was seen in both men and women [142]. 

An inverse association between blood Cd and BMI was noted in Korean men, 40–70 years of age [143]. This Korean population study observed also an inverse correlation between fasting blood glucose and Cd excretion levels, and urinary Cd levels > 2 μg/g creatinine were associated with a 1.81-fold increase in risk of diabetes.

In a Thai population study, increases in risk of diabetes in men and women were not associated with obesity/overweight, but were associated with blood Cd and lead (Pb) levels above median values of 0.3 µg/L and 2.12 µg/dL for Cd and Pb, respectively [144]. 

In a Chinese study, Cd excretion levels ≥ 2.95 µg/g creatinine were associated with reduced risk of weight gain and obesity [145]. In a study of non-occupationally exposed residents of Shanghai, the median urinary Cd excretion was 0.77 μg/g creatinine and higher urinary Cd levels were associated with lower BMI values [146]. 

### 3.3. The U.S. Population Risk Analysis of Cd-Associated Diabetes 

The geometric mean, the 50th, 75th, 90th, and 95th percentile values for urinary Cd levels in the representative U.S. general population were 0.210, 0.208, 0.412, 0.678, and 0.949 µg/g creatinine, and the corresponding values for blood Cd were 0.304, 0.300, 0.500, 1.10, and 1.60 µg/L, respectively [147]. Although the U.S. population mean Cd excretion was less than 0.5 µg/g creatinine, 2.5%, 7.1%, and 16% of non-smoking women (aged ≥20 years) were found to have Cd excretion levels > 1, >0.7, and >0.5 μg/g creatinine, respectively [148]. Given that Cd excretion > 0.5 μg/g creatinine were found in 16% of non-smoking U.S. women [148], the proportion of people, especially women, who were at risk of Cd-associated adverse effects is not negligible

Risk analysis of data from 4530 adults enrolled in NHANES 1999–2006 showed that mean Cd excretion levels of 0.198 and 0.365 μg/g creatinine were associated with the likelihood that the prevalence of diabetes to be less than 5% and 10%, respectively [115]. These Cd excretion levels were 3.78% and 6.97% of the nephrotoxicity threshold level determined from the β_2_M endpoint [107]. 

The Cd excretion of 0.198 and 0.365 μg/g creatinine associated with 5% and 10% prevalence of diabetes were in ranges with the 50th and 75th percentiles of Cd excretion. Thus, the proportion of the U.S. adults at risk of Cd-associated diabetes was substantial. Similarly, Cd excretion was inversely associated with bone mineral density, and low environmental Cd exposure in the U.S. accounted for 16% of osteoporosis cases, aged 50–79 years [118].

## 4. Cadmium, the Liver, Kidney, and Diabetes Type 2

The kidney is the only organ other than the liver that produces and releases glucose into the circulation [20,21,22,23]. The liver and kidney are directly involved in blood glucose control. The kidney contributes to 20–25% of plasma glucose after an overnight fast, and it releases into the circulation 60% of plasma glucose in the postprandial period [22]. In diabetes type 1, there is an impairment in renal release of glucose [22].

The kidney is also responsible for filtration, and reabsorption of glucose. In normal health, an approximate of 160 to 180 g of glucose is retrieved each day [20,21,22,23]. The sodium glucose co-transporter 2 (SGLT2) and SGLT1 mediate 90% and 10% of the tubular reabsorption of glucose, respectively [23]. Increased renal expression of these glucose transporters has been implicated in an elevation of renal threshold for glucose excretion in diabetes type 2 patients [21].

Loss of tubular gluconeogenesis and a switch to glycolysis are known pathologic features of CKD [23]. In clinical trials, SGLT2 inhibitors were effective to attenuate the deterioration of kidney function in CKD patients [24]. 

Cd-associated GFR reductions, albuminuria, and hypertension were more severe in those who had diabetes [129,145,149,150,151]. These results are replicated in experimental studies [152,153]. In cross-sectional and prospective cohort studies of 231 diabetic patients in the Netherlands, both Cd and active smoking were associated with a progressive decline in eGFR [154,155]. Collectively, these findings support the premise that exposure to even low levels of environmental Cd promote the development and progression of DKD.

Figure 3 depicts air and foods as sources of environmental Cd that gains access to the systemic circulation through lungs and the gastrointestinal tract.

As detailed in Section 2.4, the proximal tubule accumulates most Cd acquired, and Cd amount in kidney cortex, as µg/g wet tissue weight, is the highest (Table 1). For instance, respective mean Cd levels in lung, liver and kidney cortex of Australians, aged 2–70 years (mean 39.9) were 0.12, 0.99, and 20.5 μg/g wet tissue weight, while the mean urinary Cd was 0.62 μg/L, range; 0.05–2.88 μg/L [51]. 

In a study of kidney transplant donors, a Cd excretion of 0.42 μg/g creatinine corresponded to kidney cortical Cd of 25 μg/g wet kidney weight [156]. In female kidney transplant donors, the mean values for Cd excretion, blood Cd and kidney cortical Cd were 0.34 μg/g creatinine, 0.54 μg/L and 17.1 μg/g kidney wet weight, respectively [157]. The corresponding figures in men were 0.23 μg/g creatinine, 0.46 μg/L and 12.5 μg/g, all of which were lower than in women [157].

The rates of Cd accumulation found in Australian autopsy study were 3–5 µg/g wet tissue weight for each 10-year increase in age, reaching 25.9 µg/g wet tissue weight in 50 years [51]. After adjustment for age and inhalational exposure, the rate of Cd accumulation in kidneys was higher in females than males [51]. 

Similarly, the rate of Cd accumulation found in non-smoking Swedish kidney transplant donors was 3.9 μg/g kidney wet weight for every 10-year increase in age. Non-smoking women with low body iron stores had a Cd accumulation rate of 4.5 μg/g kidney wet weight in 10 years [53]. 

## 5. Cadmium and Diabetes: Experimental Studies 

This section provides a summary of findings from experimental studies attempted to shed light on how Cd causes diabetes. However, most experimental studies examined Cd-induced diabetes along with the impacts of high-fat diet in the belief that obesity is a major contributing factor. Also, many studies examined other suspected diabetogenic substances; polyfluoroalkyl substances [158,159], and polychlorinated biphenyls [160]. Because these chemicals are ubiquitous in the environment, co-exposure of Cd with these chemicals is a likely scenario. 

Apparently, Cd induced diabetes by multiple mechanisms. As depicted in Figure 1, Cd is a mitochondrial toxicant that induces oxidative stress, inflammation [160,161,162], disrupt ATP and intermediary metabolism, and insulin resistance in many tissues, including insulin-dependent and non-dependent types [163,164,165]. 

Furthermore, Cd may have an indirect effect on diabetes through induction of hyperuricemia. An association between prevalence of hyperuricemia and Cd exposure has been noted in Chinese [166], U.S. [167], and Korean [168,169] population studies. Pancreatic β-cell death and a reduction in glucose-stimulated insulin secretion have been demonstrated in mice with hyperuricemia due to uricase deficiency [170,171].

### 5.1. Cadmium-Induced Hyperglycemia: Landmark Observation

The ability of Cd to induce hyperglycemia was first demonstrated in neonatal rats [15] and then in adult rats [16,17]. The liver of Cd-exposed neonatal rats had a reduced glycogen, and an enhanced gluconeogenesis, evident from activity of enzymes in gluconeogenesis; pyruvate carboxylase, phosphoenolpyruvate carboxy kinase, fructose-1,6-biphosphatase, and glucose-6-phosphatase [15]. Hyperglycemia in Cd-exposed rats developed long before the onset of kidney toxicity [172]. 

In rats exposed to Cd via intraperitoneal injection for 45 days, depletion of hepatic and renal glycogen was noted along with enhanced activity of the rate-limiting enzymes in gluconeogenesis [18]. These changes remained 4 weeks after exposure cessation. 

In another study using rats, the effects of Cd on hepatic glucose metabolism remained one year after Cd treatment was discontinued [19]. The persistent Cd effects can be expected because most Cd is retained by cells, which provide ample of opportunity for Cd to exert toxicity. A study in rats showed Cd was excreted only when cell die [173]. 

Cd exposure in utero has been investigated in rats, where dams were exposed to Cd for a period of 21 days before mating, 21 days during gestation, and 21 days during lactation [174]. The effects of maternal Cd exposure on the metabolism of glucose and lipids in offsprings were examined at 21, 26, and 60 days of age. Collective data indicated changes in glucose homeostasis in pubs born from Cd-exposed dams that may increase susceptibility to development of diabetes [174]. Effects of early life exposure to Cd on the development of diabetes later in life have been reviewed by Saedi et al. [175].

### 5.2. Female Preponderance Effects of Cadmium

An effect of gender on Cd toxicity outcomes, indicated by deranged glucose metabolism has been investigated in rats exposed to Cd in drinking water for 3 months. An increment of plasma insulin levels in response to fasting and glucose stimulation due to impaired hepatic extraction of insulin was found in female rats only [176]. 

In previous studies, the gender differences in Cd toxicity have been attributed to role of female sex hormones such as progesterone and β-estradiol [177,178,179,180]. The hepatoxicity of Cd was increased in male Fischer 344 (F344) rats treated with progesterone [177,178,179]. Subsequent studies implicated the role of progesterone in an enhanced cellular Cd accumulation [180], possibly through suppression of the ZnT1 metal transporter that mediated efflux of Cd [181]. 

### 5.3. The Molecular Basis for Deranged Cellular Glucose Metabolism after Cd Exposure 

Intracellular levels of the natural cyclic AMP antagonist prostaglandyl-inositol cyclic phosphate (cyclic PIP) and cyclic adenosine monophosphate (cAMP) have been postulated as the key players in the development of insulin resistance [182,183]. Metformin, an anti-diabetic medication, has been found to have the ability to stimulate synthesis of cyclic PIP [183]. The increment of cyclic PIP by metformin appeared to account for its therapeutic actions, including the lowering of blood glucose levels, the inhibiting cAMP synthesis and gluconeogenesis, and increasing sensitivity to insulin [183]. Accordingly, it has been postulated that insulin resistance is a result of an imbalance action of cyclic PIP and cAMP [182]. 

Effects of metformin were investigated, using male Wistar rats, treated with Cd in drinking water at 32.5 ppm concentration only or Cd plus metformin (200 mg/kg/day) [184]. Cd was found to induce hyperinsulinemia, insulin resistance, adipocyte dysfunction, and loss of hepatic insulin sensitivity. Increased lipid accumulation was also seen in various tissues, while glycogen in the liver, heart, and renal cortex was diminished, but was increased in the muscle. Metformin showed a limited therapeutic efficiency on Cd-induced glucose tolerance and lipid accumulation.

Because changes in hepatic glucose metabolism in Cd-exposed rats coincided with a marked increase in synthesis of cAMP [18], the inefficacy of metformin reported by Sarmiento-Ortega et al. [184] may be due to the Cd stimulatory effects on cAMP formation and gluconeogenesis exceeded the inhibitory actions of metformin. 

### 5.4. Cadmium and Pancreatic β Cells

In cell culture, pancreatic β cells progressively accumulated Cd from medium containing Cd in nanomolar concentrations similar to human plasma Cd levels. An effect of Cd on insulin secretion occurred at the onset of cell death [185].

In another study using the human β cell line (the INS-1), Cd concentration ten-fold below the level causing cell death produced no effects on mitochondrial function, assessed with the energy charge and ATP synthesis [186]. This Cd concentration, however, induced mitochondrial morphological change toward circularity, indicative of fission. The increased circularity suggested mitochondrial adaptive response to low-level Cd. 

Thus, a sublethal Cd dose caused mitochondria to undergo morphological adaptative change as the mechanism to offset an effect of Cd on energy output and insulin secretion [186]. If cellular Cd influx continues, impairment of this organelle may contribute to cellular dysfunction and decreased viability of β-cells. 

Through mathematical modeling of oral glucose tolerance test data, an effect of Cd on the sensitivity of pancreatic β cells to glucose has also been demonstrated. Perinatal exposure to low-level Cd in mother’s milk reduced pancreatic β-cell sensitivity to glucose stimulation [187]. 

In rats, fasting plasma glucose was increased 12 weeks after Cd treatment, at which time pancreatic islets from the Cd-treated group showed less glucose-stimulated insulin release than islets from saline-treated control animals [188]. At this stage, Cd accumulation in isolated islets was 5 times higher than in pancreatic parenchyma but 30% lower than in renal cortex [188]. These relative pancreatic and renal Cd accumulation levels paralleled human data (Table 1). 

### 5.5. Cadmium and “Metal Stressed” Fat Cells

In a Swiss autopsy study, Cd was fund to accumulate in omentum visceral and abdominal subcutaneous fat tissues [189]. The adipose-derived human mesenchymal stem cells exposed to the same Cd levels found in those postmortem fat tissue caused a disruption in cellular zinc homeostasis and an increase in expression of various pro-inflammatory cytokines [189]. 

In a Spanish cohort study, Cd levels in fat tissues were higher in those with lower BMI values [190]. This observation is in addition to insulin resistance and higher plasma insulin levels in smokers with adipose tissue Cd levels in the middle tertile, compared to those with adipose tissue Cd levels in the bottom tertile [58]. 

In Cd-treated mice, abnormal differentiation of the adipocyte was evident from its small size, and a reduced secretion of adiponectin [191,192]. In Cd-treated rats, subcutaneous fat tissue accumulated more Cd than did abdominal and retroperitoneal adipose tissues, and all three fat tissue types had reduced adiponectin and leptin transcript levels [193].

The above human and experimental animal data clearly indicate the impact of Cd on function of fat cells. 

## 6. Heme Oxygenase, Cadmium, Cellular Stress Response and Defense 

This section focuses on HO-1 and HO-2 and their role in heme degradation, the maintenance of blood glucose, and cellular defense against oxidative stress. The induction of HO-1 by Cd and its consequential effects on glucose metabolism and manifestation of the cytotoxicity of Cd are highlighted. 

### 6.1. Indispensable Role of Heme Oxygenase 

HO-1 and HO-2 are proteins with a molecular weight of 32 kDa; HO-1 is known also as the heat shock protein 32 (HSP32) [194,195,196,197]. In concert with NADPH-cytochrome P450 reductase, HO-1 and HO-2 break down heme with the resultant release of Fe, CO, and biliverdin IXα [196,197,198,199]. Biliverdin IXα is converted to bilirubin almost instantly by biliverdin reductase. The bulk of Fe release by HO-1 and HO-2 is reutilized in the synthesis of hemoproteins, including nitric oxide synthase, various enzymes of the mitochondrial respiratory chain and the cytochrome P450 super family [198,199]. Two other products of heme degradation, namely CO and bilirubin, are known for their anti-inflammatory and antioxidant properties [199,200,201,202]. 

Bilirubin is lipophilic, as such it acts as a lipid peroxidation chain breaker that protects lipids from oxidation more effectively than the water-soluble antioxidants, such as glutathione [201,202,203,204]. Bilirubin contributes mostly to the total antioxidant capacity of blood plasma [204]. Apparently, heme degradation by HO-1 and HO-2 is indispensable. 

### 6.2. Heme Oxygenase Activity and Blood Glucose Levels 

Because CO is produced exclusively by HO-1 and HO-2, an exhaled CO can serve as a biomarker for heme degradation. In healthy individuals, levels of exhaled CO increase with increasing blood glucose and both exhaled CO and blood glucose levels return to their respective baseline values 40 min after glucose administration. These data suggest that levels of HO activity may influence blood glucose levels [205]. The relationship between exhaled CO and blood glucose has been observed as well in diabetics. As expected, the levels of exhaled CO are greater in diabetic subjects, compared to non-diabetic controls [205]. The elevated CO exhaled in diabetic subjects is attributable to HO-1 induction in response to high-glucose stress. The exhaled CO-blood glucose correlation implies that exhaled CO can be used in monitoring disease progression in diabetes patients [205]. 

In the Goto-Kakizaki rats, a model for hyperglycemia and insulin resistance without obesity [206], induction of HO-1 causes a reduction in fasting blood glucose levels and prevents a rise in blood glucose in post absorptive state [207]. In an obese mouse model of diabetes, induction of HO-1 prevents weight gain, decreases visceral and subcutaneous fat content, and improves both insulin sensitivity and glucose tolerance [208].

### 6.3. Similarities versus Differences between HO-1 and HO-2

The catalytic domains of HO-1 and HO-2 are highly homologous, sharing 93% of their amino acid sequences. HO-2, however, contains an additional domain, which has Cys-Pro dipeptide motifs that allows binding of heme and interacting with other proteins that include Rev-erbα, a heme sensor that coordinates metabolic and circadian pathways [209,210,211] and PFKFB4, the key regulator of glycolysis. This HO-2 domain accounts for the its biologic roles that are distinct from those of HO-1, such as protection against ischemic acute kidney injury [212] and anti-diabetic properties, detailed below.

HO-1 and HO-2 are products of two different genes; the promoter of the human HO-1 gene is unique because it contains the GT repeats, not found in rodent or murine species [194,195]. The genetic polymorphisms, such as long GT repeats, are associated with an elevated risk for various diseases, type 2 diabetes included [213,214]. 

Expression of the HO-1 gene is regulated by a cascade of transcription factors such as CLOCK, Bmal, and Per, that generate day-night cyclical expression of the genes involved in energy metabolism [215,216,217,218,219,220]. Disruption of the diurnal cycle caused obesity in mice [218]. Expression of the HO-1 gene is regulated also by heme (its own substrate), the levels of glucose, oxygen, and shear stress [200] and it is a component of innate immune responses [221].

HO-2 deficiency causes neither lethality nor infertility. The HO-2 knockout mice reproduce offsprings that undergo normal development to adulthood, but develops the symptomatic spectrum of human type-2 diabetes; hyperglycemia, increased fat deposition, insulin resistance and hypertension with aging [208,222,223,224]. Normal fertility and normal development of HO-2 knockout mice suggests that HO-1 could compensate for heme degradation function of HO-2. However, HO-1 could not compensate for anti-diabetogenic function of HO-2, thereby suggesting such function is unique to HO-2. 

### 6.4. Cellular Stress Response and Defense against Cadmium Toxicity 

Activation and repression of the HO-1 gene are universal cellular stress responses and defenses that are required for cell survival under the influence of environmental stressors of various forms. Every nucleated cell in the body must synthesize heme, and the de novo heme biosynthesis supplies heme that can rapidly be catabolized to a precursor substrate (biliverdin IXα) to produce bilirubin [198,199,200,201,202,203].

However, unlike endogenous (physiologic) HO-1 activators, the HO-1 induction by Cd is not coupled with bilirubin synthesis [225]. The rapid and massive HO-1 expression in response to Cd, leads to a transient increase in the intracellular heme concentration. This results in the stimulation of gluconeogenesis, and a shift to dominant glycolysis, a known pathologic feature of CKD [23]. 

The finding that Cd induces HO-1 expression without a concomitant bilirubin formation is of significance in Cd research. It explains the pervasiveness of Cd toxicity as it increases cellular oxidative stress and lowers cellular antioxidant capacity at the same time. This knowledge comes from a methodology breakthrough in measuring bilirubin as it is produced in cells [225,226]. Using the eel fluorescent protein UnaG, which binds unconjugated bilirubin [226], Takeda et al. (2015) demonstrated, for the first time, that all cell types that they examined synthesized heme, from which bilirubin was continuously generated and released [225]. This de novo synthesis of heme was mandatory for cellular homeostasis, and defense against stress. Takeda et al. (2015) reported also that stressors like Cd^2+^ and inorganic arsenic as As^3+^ increased HO-1 expression, but there was only a small change in the production of bilirubin. 

### 6.5. How Does Cd Activate HO-1 Expression?

The expression of HO-1 can be increased by various chemicals of endogenous and exogenous origin. Although induction of Cd by exogenous chemicals were extensively investigated, there was little knowledge on the mechanism underlying activation of the HO-1 gene by endogenous chemicals, notably prostaglandin (PGD_2_) [227,228,229]. 

PGD_2_ is a major cyclooxygenase mediator, synthesized by activated mast cells and other immune cells, and is implicated in allergic disorders [229]. In a study using a cell culture model of human retinal epithelial cells and the reporter gene assay, Satarug et al. (2008) found, for the first time, that PGD_2_ activated the HO-1 gene, in an enhancer manner, through D-prostanoid 2 (DP_2_) receptor [227]. The DP_2_ receptor is also known as a chemoattractant receptor-homologous molecule expressed on Th_2_ cells.

In comparison, Cd was found to activate HO-1 promoter via the Cd response element (*CdRE*), and Maf recognition antioxidant response element (*MARE*), also known as a stress response element (*StRE*) [230]. Cd also suppresses lysosomal degradation of Nrf2 [231] and causes nuclear export of the HO-1 gene repressor Bach1, which allows transactivation of the HO-1 gene by the Nrf2/small Maf complex [232].

### 6.6. Maintenance of Blood Glucose: Integrative Role of HO-1, HO-2 and PFKFB4 

Based on protein microarray data, Li et al. (2012) observed HO-2 interaction with 6-phosphofructo-2-kinase/fructose-2,6-biphosphatase 4 (PFKFB4), thereby linking HO-2 to glylcolysis [224]. In liver, PFKFB4 is the key regulator of glycolysis [233] and HO-2 deficiency causes persistent hyperglycemia due to an impaired ability to suppress glucose production. 

PFKFB4 expression is regulated by the hepatocyte nuclear factor-6 (Hnf-6) [234] and diabetes developed in Hnf6-knockout mice [235]. PFKFB4 protein phosphorylation, mediated by the cAMP dependent protein kinase A (PKA) reduced F-2,6-P_2_ level in the liver, thereby increasing gluconeogenesis with concomitantly reducing glycolysis [236]. Figure 4 depicts the regulation of blood glucose in fasting and post absorptive states. 

Both HO-1 and HO-2 are required to prevent a fall or a rise in blood glucose levels during fasting and post absorptive periods, respectively. In fasting state, HO-1 up-regulation concurrent with PFKFB4 down-regulation results in enhanced glucose production with minimal use of glucose. In the post-absorptive state, HO-1 down regulation concurrent with HO-2 plus PFKFB4 up-regulation results in suppressed glucose production and increased use of glucose. 

HO-2 is required for the up-regulation of PFKFB4. Failure in any of these (HO-1, HO-2 or PFKFB4) can result in hyperglycemia due to over production of glucose in fasting state in combination with an impaired ability to suppress its production.

HO-1 protein expression in the liver of HO-2 deficient mice was lower than the wild type by 35–45% [223,235]. This markedly low HO-1 expression level could render the hepatocyte to oxidative damage. However, the repression of the HO-1 gene expression is a necessary metabolic adaptation to safeguard the cellular redox state. This could be achieved by utilizing NADPH (H+) for regenerating GSH from GSSG, an oxidized form of GSH, rather than for heme catabolism. GSH recycling is a mechanism for maintenance of cell redox state. It is central to cell function integrity.

## 7. Conclusions

Even a small increase in the risk of diabetes by Cd exposure yields a large number of cases that are preventable by early minimization of exposure. Current dietary exposure guidelines and a nephrotoxicity threshold of Cd do not afford health protection. Cd is a mitochondrial toxicant that induces, in multiple tissues and organs, oxidative stress, chronic systemic inflammation, and insulin resistance independently of adiposity. Cd has a high toxicity potential because it induces oxidative stress and reduces cellular defense and antioxidant capacity, simultaneously. These effects of Cd may be intensified in obese persons. Thus, the risk of diabetes is higher in the obese, compared to the non-obese with the same overall Cd burden. 

Cd induces HO-1 expression without a concomitant increase in bilirubin synthesis, but stimulates gluconeogenesis, leading to hyperglycemia. Metformin is ineffective to prevent the expression of diabetic symptoms induced by Cd. 

Minimization of Cd exposure from all sources are essentially preventive measures. Adequate Zn and Fe intake and maintaining optimal body Fe stores are additional interventions. An increase in endogenous bilirubin production may be a complementary measure to mitigate harmful effects of inevitable exposure to Cd. 

Further research dissecting the molecular basis for a renoprotection of HO-2 and its anti-obese and anti-diabetogenic properties are imperative. 

## Figures and Tables

**Figure 1 cells-13-00083-f001:**
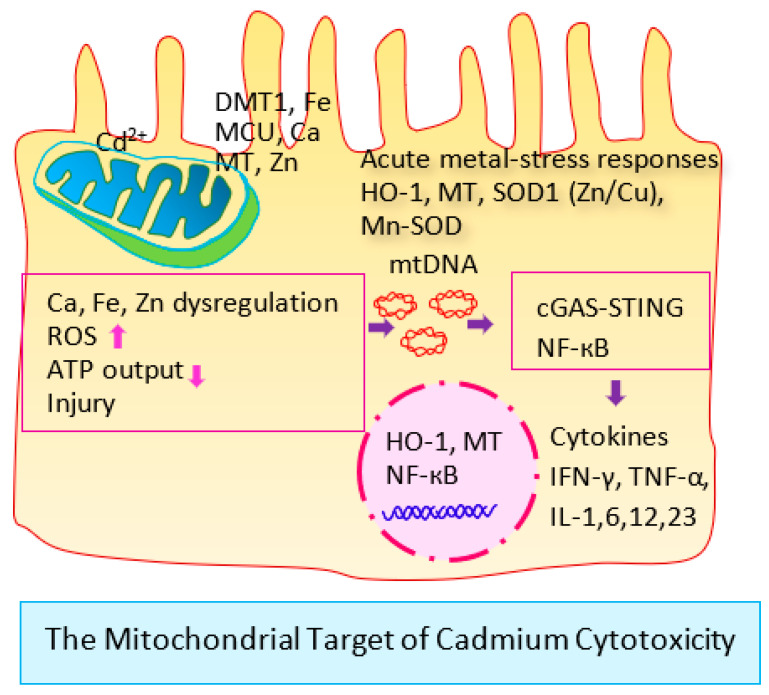
Kidney tubular cell injury and death after exposure to cadmium. Cd reaches the inner membrane of mitochondria through the metallothionein (MT) and transporters of Ca and Fe, metal coupling unit (MCU) and the divalent metal transporter1 (DMT1) [14]. There, Cd induces dysregulation of Ca, Fe and Zn, reduces synthesis of ATP (↓), promotes (↑) formation of reactive oxygen species (ROS), and mitochondrial injury. Consequently, mitochondrial DNA (mtDNA) is released, leading to activation of the DNA-sensing mechanism (cGAS-STING) and nuclear factor-kappaB (NF-κB) signaling pathways, a release of proinflammatory cytokinesand cell death.

**Figure 2 cells-13-00083-f002:**
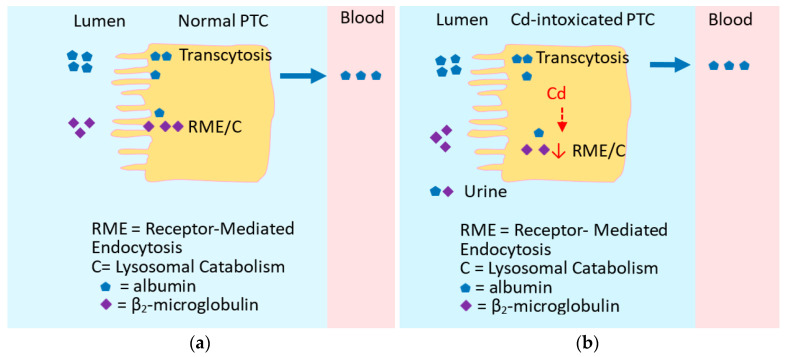
Protein re-absorption by the kidney proximal tubular cells (PTCs). (**a**) Reabsorption of albumin and β_2_-microglobulin (β_2_M) in lumen through transcytosis and receptor-mediated endocytosis (RME). (**b**) Cd-induced RME dysfunction compromises reabsorption, and increases excretion of albumin and β_2_M. In Cd-intoxicated PTCs, unbound Cd may impair the function of RME (↓).

**Figure 3 cells-13-00083-f003:**
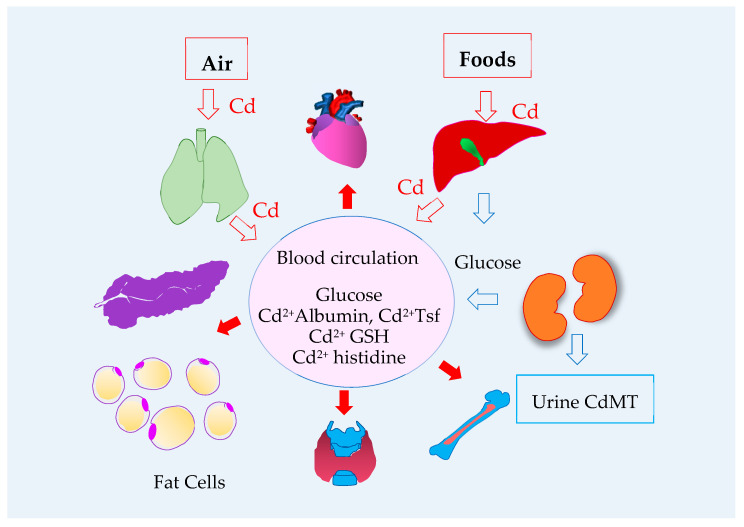
Sources, entry routes and systemic transport of cadmium. Inhaled Cd enters the systemic circulation through lungs. Cd absorbed from foods is transported to liver via the portal blood system, and is taken up by hepatocytes. The absorbed Cd not taken up by hepatocytes in the first pass enters the systemic circulation and reaches tissues and organs throughout the body, including the heart, kidney, bone, thyroid gland, fat cells and pancreas. The kidney and the liver are the only two organs that produces and releases glucose into the circulation. Due to toxic Cd accumulation, kidney tubular cells die, and Cd complexed with metallothionein (CdMT) are released into tubular lumen and excreted. Thus, excreted Cd signifies the toxicity of Cd in tubular cells. Abbreviations: Cd = cadmium; GSH = glutathione; Tsf = transferrin.

**Figure 4 cells-13-00083-f004:**
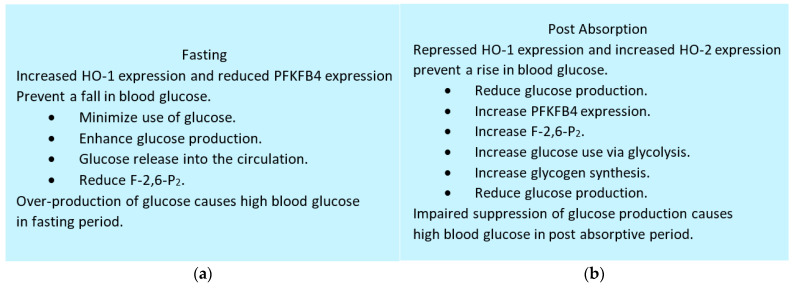
Regulation of blood glucose by hepatic HO-1, HO-2 and PFKFB4 (**a**) Expression of HO-1 and PFKFB4 in the fasting state.; (**b**) Expression of HO-1, HO-2 and PFKFB4 in the post absorptive period. Abbreviations: PFKFB4, 6-phosphofructo-2-kinase/fructose-2,6-biphosphatase 4; F-2,6-P_2_, fructose 2,6-biphosphate.

**Table 1 cells-13-00083-t001:** Gender- and organ-differentiated cadmium accumulation in Australia and Japan citizens.

Tissues/Organs	Cd Content in µg/g Wet Tissue Weight		Country of Origin/Reference
	Males	Females	
Lung	0.11 ± 0.19	0.17 ± 0.35	Australia, Satarug et al. [51] ^a^
Liver	0.78 ± 0.71	1.36 ± 0.96	
Kidney cortex	14.6 ± 12.4	18.1 ± 18.0	
Liver	7.9 (1.3−33.3)	13.1 (3.1−106.4)	Japan, Uetani et al. [52] ^b^
Kidney cortex	72.1 (19.4−200)	83.9 (3.9−252.9)	
Kidney medulla	18.3 (3.5−76.4)	24.5 (4−105)	
Pancreas	7.4 (3.0−25.9)	10.5 (2.5−29.8)	
Thyroid	10.6 (3.8−35)	11.9 (3.9−56.4)	
Heart	0.3 (0.1−0.5)	0.4 (0.1−1.3)	
Muscle	1.2 (0.3−3.2)	2.2 (0.8−12.4)	
Aorta	1.0 (0.4−2.5)	1.1 (0.3−3)	
Bone	0.4 (0.2−0.6)	0.6 (0.2−1.6)	

^a^ Values were arithmetic mean ± standard deviation from 43 males and 18 females, aged 2–89 years (mean 38.5). ^b^ Values were geometric mean (lowest−highest Cd levels) from 36 males and 36 females, aged 60–91 years (mean 74).

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
