# Peer review of "Is Environmental Cadmium Exposure Causally Related to Diabetes and Obesity?"

_cells, 2023, doi:10.3390/cells13010083_

Round 1
Reviewer 1 Report
Comments and Suggestions for Authors
This is an interesting review with very richen information. However, this structural and conceptual contents should be improved with more clear rationales, explainable links, and discussion of the descripting outcomes with certain hypotheses, to avoid the literature pilling.
Majors
1. The title is interesting even though it is not novel. However, the body of reviewed contents were too wide to clearly see the logical links to this topic. Particularly the Section of 2 was too loos without a clear summary (about 5 full pages), which should be briefly introduced with well-organized information that will useful for the following other sections (less than 2-3 pages).
2. Sections 3 and 4, should be combined into one section “Association of Cd with diabetes” by (1) removing the association of Cd with CVDs that are diabetic complications, instead of causative factors to diabetes”; (2) combined the positive association of Cd levels with obesity, insulin resistance and prediabetes as one part; (3) emphasizing the positive association of Cd with diabetes (including type 1 and type 2 diabetes; (4) removing off liver disease and kidney disease since the potential contribution of kidney and liver could be discussed in the section of 6 Potential mechanisms responsible for Cd contribution to diabetes; (5) Negative associations of Cd levels with diabetes should be discussed as contradictory results whether certain conditions should be considered, instead of simply piling as evidence.
3. Section of 6 is better use the subtitle something such as “Potential mechanisms responsible for Cd contribution to diabetes” to discuss the potential contributions of Cd-mediated various effects including insulin/beta cells effects, glucose impact by liver and kidney as well as molecular glucose metabolic pathways (major information of current section 6, but need to be synthesized well).
Minors
4. Lines 61 – 63: This statement is important, but lacks citation of references to support.
5. Lines 65- 68. The prediabetes and diabetes diagnosis should be based on those provided by either WHO or American Diabetes Association with which should be cited since this is a key point for this review.
Author Response
Thank you for reviewing my work. Please see the attachment.

Reviewer 2 Report
Comments and Suggestions for Authors
To further strengthen the study, it's recommended that the authors indicate elucidating the relationship and mechanisms underlying cadmium toxicity and diabetes. One proposed enhancement involves incorporating additional data from animal studies where cadmium is deliberately administered to induce diabetes. By doing so, the study could substantiate whether cadmium plays a direct role or acts as a significant risk factor for diabetes. This would fortify the link between cadmium exposure and the onset of diabetes, offering a more comprehensive understanding of its impact.
Abstract
Major comments
The abstract ought to present a coherent narrative by interconnecting all the provided data. Authors must establish a seamless flow that links various data points, ensuring a cohesive representation of their study. Additionally, authors should consider concluding the abstract with a concise summary that encapsulates the key findings or insights derived from the research.
Minor comments
Line 13: the word here should be removed from the sentence.
Line 15: The author seems to jump from one subject to another. It needs a sentence that connects two subjects.
Introduction
Minor comments
Line 39-40: need a reference
Line 54-55: please rephrase it.
Line 64-66: The authors indicate the aim of the study in this line, it should move at the end of the introduction.
Line 74: need reference
Line 75: As a suggestion, please add the following reference: Babazadeh, D., Shabestari Asl, A., Sadeghi, A., Saeed, M., & Moshavery, A. (2022). Comparative Histopathologic Evaluation of the Effects of Portulaca oleracea, Omega-3, and Combination of Sodium Selenite and Vitamin E on Hepatic Enzymes of Experimental Diabetic Rats. Small Animal Advances, 1(1), 4–9. https://doi.org/10.58803/saa.v1i1.2
Line 83: Authors should provide the aim and novelty of the study.
Exposure Sources, Dosimetry, and Health Risk Assessment
Figure 3 can be redesigned and has more details, especially on the left side of the image, the meaning of the images is not clear.
Line 87: Need reference.
Line 108-111: need a reference.
Line 115: Authors indicate the contribution of exposure to Cd in rice and its products, green vegetables, cereals, seeds, and potatoes but they indicate only 3 percent of contribution exposure.
Line 146: Please indicate Cd accumulation levels of tissues, recorded in other countries (not only Japan and Australia).
Line 148: Please indicate the countries in the title that used their data in the table.
Line 148: In Table 1 authors indicated the year of study for one reference but did not indicate for another reference.
Line 164: the subtitle 2.3 should be reconsidered. The authors mentioned “Blood and plasma (serum) cadmium”. If authors indicate blood cadmium, it would be clearer.
Line 190-192: Please provide more explanation and indicate a reference.
Line 212: Please provide a figure with a higher resolution.
Line 283: Authors should indicate the country of each reference.
Line 316-320: needs references.
Line 335: Please correct the language error “increase not in-creased”
Line 347: Please define Pb
Line 421: The authors indicate most experimental studies examined Cd-induced diabetes along with the impacts of a high-fat diet what about diabetes with other causes?
Line 478: Please change the figure to a high-resolution figure.
Line 484-497: what is the relation of these paragraphs with the rest of the study?
Line 536: Please indicate the species. Effects of Cd on hepatic glucose metabolism in what species?
References
There are many old references in the study that the author should replace with recent studies.
DOI should be indicated for all references.
Comments on the Quality of English LanguageThe article is well written and just a few phrases should be checked by the editor.
Author Response
Reviewer 1
Comments and Suggestions
To further strengthen the study, it's recommended that the authors indicate elucidating the relationship and mechanisms underlying cadmium toxicity and diabetes. One proposed enhancement involves incorporating additional data from animal studies where cadmium is deliberately administered to induce diabetes. By doing so, the study could substantiate whether cadmium plays a direct role or acts as a significant risk factor for diabetes. This would fortify the link between cadmium exposure and the onset of diabetes, offering a more comprehensive understanding of its impact.
RESPONSE:
- Thank you for evaluating this work, comments, suggestions, and guidance to improve a manuscript.
- Accordingly, this manuscript has undergone extensive revisions and topic/subtopic reorganization. Changes to the text in a manuscript are in blue
- I disagree with the reviewer’s remark that additional data from animal studies showing diabetogenic action of Cd are required.
- In the original submission, an entire Sections 5 Cadmium and Diabetes: Experimental Studies is devoted for Cd-induced diabetes in experimental animals. Further information on diabetogenicity of Cd is provided in Section 6.3. Cadmium-induced hyperglycemia and Section 6.4. The molecular basis for deranged cellular glucose metabolism after Cd exposure.
- Former Sections 6.3 and 6.4 have now been merged with Section 5 to unified all experimental studies.
- From experimental data, there is little doubt that Cd is diabetogenic. The question remains is how does Cd cause diabetes?
- Comment on Abstract
Major comments
The abstract ought to present a coherent narrative by interconnecting all the provided data. Authors must establish a seamless flow that links various data points, ensuring a cohesive representation of their study. Additionally, authors should consider concluding the abstract with a concise summary that encapsulates the key findings or insights derived from the research.
Minor comments
Line 13: the word here should be removed from the sentence.
Line 15: The author seems to jump from one subject to another. It needs a sentence that connects two subjects.
RESPONSE: Abstract has been rewritten, where the fact that Cd is a diabetogenic substance and protective roles of HO-1 and HO-2 are reflected (lines 8-27). The issues pertaining to lines 13 and 15 have been addressed.
- Comment on Introduction
Minor comments
2.1. Line 39-40: need a reference
2.2. Line 54-55: please rephrase it.
2.3 Line 64-66: The authors indicate the aim of the study in this line, it should move at the end of the introduction.
2.4. Line 74: need reference
RESPONSE:
- Introduction has been rewritten and aims are explicitly stated in the last paragraphs (lines 66-73).
- The comments above with regards to missing references are no longer relevant.
2.5 Line 75: As a suggestion, please add the following reference: Babazadeh, D., Shabestari Asl, A., Sadeghi, A., Saeed, M., & Moshavery, A. (2022). Comparative Histopathologic Evaluation of the Effects of Portulaca oleracea, Omega-3, and Combination of Sodium Selenite and Vitamin E on Hepatic Enzymes of Experimental Diabetic Rats. Small Animal Advances, 1(1), 4–9. https://doi.org/10.58803/saa.v1i1.2
RESPONSE:
- As I understand it, a suggested paper indicated a possible adverse effect of diabetes (regardless of the causes) on liver. It has been inserted in Section 3. Cadmium, Obesity and Diseases with High Prevalence (lines 301-302).
2.6 Line 83: Authors should provide the aim and novelty of the study.
RESPONSE:
- The purpose statements have now been placed here (the last paragraph of Introduction).
- Comment on Exposure Sources, Dosimetry, and Health Risk Assessment
3.1. Figure 3 can be redesigned and has more details, especially on the left side of the image, the meaning of the images is not clear.
RESPONSE:
- Arrows has been added to Figure 3 to connect sources with route of entry and the delivery of Cd through systemic circulation. Figure legend has been rewritten to better explain figure key elements.
3.2. Line 87: Need reference.
RESPONSE: Provided as applicable.
3.3. Line 108-111: need a reference.
RESPONSE: Provided as applicable.
3.4. Line 115: Authors indicate the contribution of exposure to Cd in rice and its products, green vegetables, cereals, seeds, and potatoes but they indicate only 3 percent of contribution exposure.
RESPONSES:
- The reviewer misread it. The statement in former line 115 reads as below.
- Respective contribution of exposure to Cd in rice and its products, green vegetables, cereals, and seeds plus potatoes were 38%, 17%, and 11%.
3.5. Line 146: Please indicate Cd accumulation levels of tissues, recorded in other countries (not only Japan and Australia).
RESPONSE:
- In the present review data from Australia and Japan are chosen because these two studies reported Cd accumulation in many organs, notably lung, liver and kidney which show a distinctly high Cd in kidney., compared with lung and liver.
- In most studies data are available for a single or a couple of organs.
- Please note, in original submission, data on renal Cd accumulation levels in Swedish kidney transplant donors have also been provided. Not only Australian and Japanese data.
3.6. Line 148: Please indicate the countries in the title that used their data in the table.
RESPONSE:
- Table 1. Gender- and organ-differentiated cadmium accumulation in Australia and Japan citizens
3.7. Line 148: In Table 1 authors indicated the year of study for one reference but did not indicate for another reference.
RESPONSE:
- For uniformity, the year of publication has been deleted.
3.8. Line 164: the subtitle 2.3 should be reconsidered. The authors mentioned “Blood and plasma (serum) cadmium”. If authors indicate blood cadmium, it would be clearer.
RESPONSE: Correction has been undertaken.
3.9. Line 190-192: Please provide more explanation and indicate a reference.
RESPONSE: Provided as applicable. Due to extensive revision, it may not be pertinent.
3.10. Line 212: Please provide a figure with a higher resolution.
RESPONSE: An editable figure has been provided.
3.11. Line 283: Authors should indicate the country of each reference.
RESPONSE:
- Data in an entire Table 2 are from the U.S. population. Thus, a table title has been changed to read as below.
- Table 2. Urinary and blood cadmium levels associated with increased risks of liver and kidney disease in the United States.
3.12. Line 316-320: needs references.
RESPONSE: Provided as applicable
3.13. Line 335: Please correct the language error “increase not in-creased”
RESPONSE: Typo error has been corrected.
3.14. Line 347: Please define Pb
RESPONSE: “Pb” has been replaced with “lead (Pb)”.
3.15. Line 421: The authors indicate most experimental studies examined Cd-induced diabetes along with the impacts of a high-fat diet what about diabetes with other causes?
RESPONSE:
- I agree that there are substances other than Cd that are suspected to be diabetogenic and responsible for the epidemic of diabetes type 2. These include per- and polyfluoroalkyl substances and polychlorinated biphenyls.
- It would be relevant to study interactions of these suspected diabetogenic chemicals
- Unfortunately, so far there is no such study.
- In revision, below statements and three new references are inserted (lines 431-433).
Furthermore, many studies examined other suspected diabetogenic substances such as polyfluoroalkyl substances [1.2] and polychlorinated biphenyls [3]. Co-exposure of Cd with these chemicals is a likely scenario because they all are ubiquitous in the environment.
[3] Jacquet, A.; Ounnas, F.; Lénon, M.; Arnaud, J.; Demeilliers, C.; Moulis, J.M. Chronic exposure to low-level cadmium in diabetes: Role of oxidative stress and comparison with polychlorinated biphenyls.
Curr. Drug Targets 2016, 17, 1385-1413.
[1] Roth, K.; Petriello, M.C. Exposure to per- and polyfluoroalkyl substances (PFAS) and type 2 diabetes risk. Front. Endocrinol (Lausanne) 2022, 13, 965384.
[2] Gui, S.Y.; Qiao, J.C.; Xu, K.X.; Li, Z.L.; Chen, Y.N.; Wu. K.J.; Jiang. Z.X.; Hu. C.Y. Association between per- and polyfluoroalkyl substances exposure and risk of diabetes: a systematic review and meta-analysis. J. Expo. Sci. Environ. Epidemiol. 2023, 33, 40-55.
3.16. Line 478: Please change the figure to a high-resolution figure.
RESPONSE: Figure 4 has been deleted.
3.17. Line 484-497: what is the relation of these paragraphs with the rest of the study?
RESPONSE:
- An entire Section 6 has been reorganized and short introductory statements are provided.
3.18. Line 536: Please indicate the species. Effects of Cd on hepatic glucose metabolism in what species?
RESPONSE:
- The animal species has been inserted.
- Comment on References
4.1. There are many old references in the study that the author should replace with recent studies.
RESPONSES:
- I disagree. With only an exception for the pioneering work in 1980s, experimental studies cited in Section 5 mostly are relatively new and considered to be of high qualities.
- It is noteworthy that most recent experimental studies on Cd and diabetes provide so little new knowledges, and sometimes is misleading.
- The ability of Cd to induce hyperglycemia and its effect on glucose metabolism were well described in 1980s (Section 5). These early works deserve recognition to advance science.
- There is a flood of information on diabetes, and numerous promises to cure this disease. It is difficult for junior researchers to judge scientific merits of published reports these days.
- To address diabetogenic action of Cd in lean persons requires suitable animal models of spontaneous diabetes such as the Goto-Kakizaki rats.
- In a study from Uganda, 3 in 4 adults with diabetes were lean (Kibirige et al. 2022) and yet most studies on the effect of Cd employed obese/obese mice or rats fed with high-fat diet.
Goto, Y.; Kakizaki, M.; Masaki, N. Production of spontaneous diabetic rats by repetition of selective breeding. Tohoku J. Exp. Med. 1976, 119, 85-90.
Kibirige, D.; Sekitoleko, I.; Lumu, W.; Jones, A.G.; Hattersley, A.T.; Smeeth, L.; Nyirenda, M.J. Understanding the pathogenesis of lean non-autoimmune diabetes in an African population with newly diagnosed diabetes. Diabetologia 2022 65, 675-683.
4.2. DOI should be indicated for all references.
RESPONSES: As I understand it, DOI can be easily added by editorial staff.
Comments on the Quality of English Language
The article is well written and just a few phrases should be checked by the editor.
RESPONSES: Thank you.
Reviewer 3 Report
Comments and Suggestions for Authors
The manuscript describes the association among Cd exposure, diabetes and obesity. In my opinion entire manuscript should be improved.
1. There are too many keywords (11).
2. Author did not analyze the influence of hormone status on the Cd concentration which is quit pivotal, especially when evaluating sex differences are (verse 158)
3. verse 153: “The mean liver Cd” concentration? level? Please clarify.
4. verse 172: Most of the circulating Cd is bound to hemoglobin in red blood cells [79]. Author cited paper from 1957 year. The abstract and the full text is not available. Did author read whole publication? This sentence should be also rewritten, because "a major amount of Cd in red blood cells is bound to high-molecular-weight proteins, while a minor amount is bound to hemoglobin"
5. verse 317: please extended with more details “various measurements of obesity”
6. In many paragraphs author did not explain the findings obtained by other authors, e.g. 323-327; 342-345; verse 344: an inverse correlation between fasting blood glucose and urinary Cd excretion levels, verse 452: fasting plasma glucose was increased 12 weeks after Cd treatment; verse 531: Hyperglycemia in Cd-exposed rats; verse 342: An inverse association between blood Cd and BMI; verse 469: Cd-treated rats, subcutaneous fat tissue accumulated more
7. verse 381: what is gm?
8. verse 422-425 the references should be introduce
9. The quality of Figure 4 should be improved
10. Conclusions section should be shorted and the main statements should be better pointed.
11. In my opinion before author analyze the relationship between Cd and HO-1 or HO-2, the characteristic of heme oxygenase enzymes should be presented
Author Response
Reviewer 2
Comments and Suggestions
The manuscript describes the association among Cd exposure, diabetes and obesity. In my opinion entire manuscript should be improved.
RESPONSE:
- Thank you for evaluating this work, comments, and suggestions to improve this manuscript.
- Accordingly, this manuscript has undergone extensive revisions and reorganization topics and subtopics. Changes to the text in a manuscript are in blue.
- Abstract and conclusion have been rewritten to better reflect the content and notable findings.
- Introduction has been rewritten and aims are explicitly stated in the last paragraphs (lines 54-72).
- More references have been added.
- There are too many keywords (11).
RESPONSES: There are now 7 keywords.
- Author did not analyze the influence of hormone status on the Cd concentration which is quite pivotal, especially when evaluating sex differences are (verse 158.)
RESPONSES:
- Thank you for raising this issue.
- I have now addressed fully this issue in Subsection 5.2. Female preponderance effects of cadmium (lines 468-478) and 5 additional references inserted.
[185] Shiraishi, N.; Barter, R.A.; Uno, H.; Waalkes, M.P. Effect of progesterone pretreatment on cadmium toxicity in the male Fischer (F344/NCr) rat. Toxicol. Appl. Pharmacol. 1993, 118, 113-118.
[186] Shimada, H.; Hochadel, J.F.; Waalkes, M.P. Progesterone pretreatment enhances cellular sensitivity to cadmium despite a marked activation of the metallothionein gene. Toxicol. Appl. Pharmacol. 1997, 142, 178-185.
[187] Shimada, H.; Hashiguchi, T.; Yasutake, A.; Waalkes, M.P.; Imamura, Y. Sexual dimorphism of cadmium-induced toxicity in rats: involvement of sex hormones. Arch. Toxicol. 2012, 86, 1475-1480.
[188] Takiguchi, M.; Cherrington, N.J.; Hartley, D.P.; Klaassen, C.D.; Waalkes, M.P. Cyproterone acetate induces a cellular tolerance to cadmium in rat liver epithelial cells involving reduced cadmium accumulation. Toxicology 2001, 165, 13-25.
[189] Ohana, E.; Sekler, I.; Kaisman, T.; Kahn, N.; Cove, J.; Silverman, W.F.; Amsterdam, A.; Hershfinkel, M. Silencing of ZnT-1 expression enhances heavy metal influx and toxicity. J. Mol. Med. 2006, 84, 753-763
- verse 153: “The mean liver Cd” concentration? level? Please clarify.
RESPONSES:
- The referred phrase has been changed (lines 152-153) as below.
- On average, the hepatic Cd level in Australian women was 1.74-fold higher than men.
- verse 172: Most of the circulating Cd is bound to hemoglobin in red blood cells [79]. Author cited paper from 1957 year. The abstract and the full text is not available. Did author read whole publication? This sentence should be also rewritten, because "a major amount of Cd in red blood cells is bound to high-molecular-weight proteins, while a minor amount is bound to hemoglobin"
RESPONSES:
- The binding of most Cd to hemoglobin in red blood cells, reported by Carlson and Friberg, 1957 is correct. This work has been reiterated in a recent review by Nordberg and Nordberg, 2022. Relevant information can be found in a paper by Gibson et al., 2017. This report was provided in the original submission.
- A full paper by Carlson and Friberg, 1957 was retrieved through the University of Queensland interlibrary services.
- I studied an entire paper as accurate information was mandatory in an attempt to publish a research article, “The pathogenesis of albuminuria in cadmium nephropathy”. This paper has now been published and cited in this review [ref. # 76].
- Three more references have been inserted (Nordberg et al. 1975; Nordberg and Nordberg, 2022; Satarug et al. 2024).
[72] Carlson, L.A.; Friberg, L. The distribution of cadmium in blood after repeated exposure. Scand. J. Clin. Lab. Invest. 1957, 9, 67-70.
[73] Nordberg, G.F.; Piscator, M.; Nordberg, M. On the distribution of cadmium in blood. Acta Pharmacol. Toxicol. 1971, 30, 289-295.
[74] Gibson, M.A.; Sarpong-Kumankomah, S.; Nehzati, S.; George, G.N., Gailer, J. Remarkable differences in the biochemical fate of Cd2+, Hg2+, CH3Hg+ and thimerosal in red blood cell lysate. Metallomics 2017, 9, 1060-1072.
[75] Nordberg, M.; Nordberg, G.F. Metallothionein and cadmium toxicology-Historical review and commentary. Biomolecules. 2022, 12, 360.
[76] Satarug, S.; Vesey, D.A.; Gobe, G.C.; Phelps, K.R. The pathogenesis of albuminuria in cadmium nephropathy. Curr. Res.Toxicol, 2024, 6, 100140.
- verse 317: please extended with more details “various measurements of obesity”
RESPONSES:
- The sentence has been changed as below.
- Studies from the U.S., and other countries consistently observed inverse relationships of urinary and blood Cd levels with various measurements of adiposity, including increases in BMI, hip girth, and waist circumference.
- In many paragraphs author did not explain the findings obtained by other authors, e.g. 323-327; 342-345; verse 344: an inverse correlation between fasting blood glucose and urinary Cd excretion levels, verse 452: fasting plasma glucose was increased 12 weeks after Cd treatment; verse 531: Hyperglycemia in Cd-exposed rats; verse 342: An inverse association between blood Cd and BMI; verse 469: Cd-treated rats, subcutaneous fat tissue accumulated more.
RESPONSES:
- These reports have provided little new information regarding mechanisms, but repletion of observations.
- I therefore have not provided further details.
- verse 381: what is gm?
RESPONSES: A typo error that has now been corrected to “g”.
- verse 422-425 the references should be introduced
RESPONSES:
- Former verse 422-425 was an introductory statement for Section 5. Cadmium and Diabetes: Experimental Studies. I made no references to those numerous studies employing obese/obese mice or rats fed with high-fat diet to study Cd effects. I considered they are not useful to the subject matter of this review (diabetogenic effects of Cd).
- For further clarification, I have made references to other suspected diabetogenic substances that have been increasingly studies, as quoted below.
- Furthermore, many studies examined other suspected diabetogenic substances such as polyfluoroalkyl substances [161.162] and polychlorinated biphenyls [163]. Because these chemicals are ubiquitous in the environment, co-exposure of Cd is a likely scenario.
- To reflect that diabetes in lean persons is an understudied area, I have added, a study from Uganda that reported 3 in 4 adults with diabetes were lean.
[161] Roth, K.; Petriello, M.C. Exposure to per- and polyfluoroalkyl substances (PFAS) and type 2 diabetes risk. Front. Endocrinol (Lausanne) 2022, 13, 965384.
[162] Gui, S.Y.; Qiao, J.C.; Xu, K.X.; Li, Z.L.; Chen, Y.N.; Wu. K.J.; Jiang. Z.X.; Hu. C.Y. Association between per- and polyfluoroalkyl substances exposure and risk of diabetes: a systematic review and meta-analysis. J. Expo. Sci. Environ. Epidemiol. 2023, 33, 40-55.
[163] Jacquet, A.; Ounnas, F.; Lénon, M.; Arnaud, J.; Demeilliers, C.; Moulis, J.M. Chronic exposure to low-level cadmium in diabetes: Role of oxidative stress and comparison with polychlorinated biphenyls.
Curr. Drug Targets 2016, 17, 1385-1413.
- The quality of Figure 4 should be improved
RESPONSES: This Figure 4 has been deleted.
- Conclusions section should be shorted and the main statements should be better pointed.
RESPONSES:
- Conclusion has been shortened to retain key elements.
- In my opinion before author analyze the relationship between Cd and HO-1 or HO-2, the characteristic of heme oxygenase enzymes should be presented
RESPONSES:
- An entire Section 6 has been reorganized and short introductory statements are provided. HO-1 and HO-2 are now better explained.

Round 2
Reviewer 1 Report
Comments and Suggestions for Authors
The authors have well addressed my concerns even though the author disagreed with my comments. The revised review after taking all comments from different reviewers reads well now. I have no further concern.
Author Response
Thank you for approving a revised version of a manuscript.
Reviewer 3 Report
Comments and Suggestions for Authors
In current version I accept the manuscript
Author Response
I am grateful to your approval of a revised manuscript.